# Patients’ Experiences and Communication with Teledermatology versus Face-to-Face Dermatology

**DOI:** 10.3390/jcm11195528

**Published:** 2022-09-21

**Authors:** Cesar Leal-Costa, Antonio Lopez-Villegas, Mercedes Perez-Heredia, Miguel Angel Baena-Lopez, Carlos Javier Hernandez-Montoya, Remedios Lopez-Liria

**Affiliations:** 1Nursing Department, University of Murcia, 30120 Murcia, Spain; 2Laboratory for Research, Education and Planning in Critical and Intensive Care Medicine, CTS-609 Research Group, Poniente University Hospital, 04700 El Ejido-Almeria, Spain; 3Research Management Department, Primary Care District Poniente of Almería, 04700 El Ejido-Almeria, Spain; 4Medical Deputy, Poniente University Hospital, 04700 El Ejido-Almeria, Spain; 5Dermatology Unit, Poniente University Hospital, 04700 El Ejido-Almeria, Spain; 6Health Research Centre, Department of Nursing, Physiotherapy and Medicine, University of Almería, La Cañada de San Urbano, 04120 Almeria, Spain

**Keywords:** dermatology, communication, eHealth, experiences, information, telemedicine

## Abstract

(1) Background: Teledermatology (TD) has exponentially grown since the onset of COVID-19, as the Face-to-Face Dermatology (F-F/D) modality changed within Public Health Systems. Although studies have been conducted on health results, we did not find any that analyzed the experiences of individuals who received care through TD. Therefore, the main objective of the study was to analyze the experiences of dermatology patients and the communication with health personnel. (2) Methods: A multicenter, controlled, randomized, non-blinded clinical trial was designed. Data were collected during the six months of follow-up. Four-hundred and fifty patients participated in the present study, who were assigned to two different groups: TD and F-F/D. The sociodemographic and clinical characteristics of the participants were collected. The ‘Generic Short Patient Experiences Questionnaire’ (GS-PEQ) was used to assess patients’ experiences, and the Healthcare Communication Questionnaire (HCCQ) was used to measure the communication of patients with healthcare professionals. (3) Results: After six months of follow-up, 450 patients completed the study (TD = 225; F-F/D = 225) of which 53.3% were women, with an average age of 52.16 (SD = 19.97). The main reasons for the consultations were skin lesions (51.56%) located on the head and neck (46.8%), followed by the legs (20.7%). According to the GS-PEQ, TD users indicated having a greater confidence in the professional skills of the doctors (*p* < 0.01). However, the F-F/D group indicated having received more adequate information about their diagnosis/afflictions (*p* < 0.01), were more involved in the decisions related to their treatment (*p* < 0.01), and more satisfied with the help and treatment received (*p* < 0.01). Regarding the HCCQ, the TD group obtained better assessments with respect to if the patients had been treated in a rude and hasty manner, if the health professionals had addressed them with a smile, and if these could adequately manage the reason for the consultation (*p* < 0.01). (4) Conclusions: The results of the study generally showed positive experiences and communication. The TD group indicated having received less information about the diagnosis, were less involved in the decisions, and were less satisfied with the help and treatment received. However, they indicated having more confidence on the professional skills of the doctors, and that the work at the institution was better organized. In addition, they perceived better communication skills of the health professionals, although less respect for their privacy.

## 1. Introduction

Skin diseases are a frequent reason for consultation and are the fourth most common cause of disease since at least one-third of the world’s population suffers from at least one dermatological condition [1,2,3,4].

Teledermatology (TD) is a type of telemedicine, which implies remote consultation of patients with skin problems with health care providers [5]. Despite the use of TD for more than 20 years, its implementation has been slow, with a rise observed in the last few years due to the COVID-19 pandemic, given the change in the face-to-face consultation scheme within health systems. TD can be implemented in a variety of primary care environments, including Primary Health Centers, hospitals, nursing homes, and rural areas without access to a dermatologist [6,7,8].

The practice of TD has evolved, given the improvement in technology, especially mobile devices [9,10] and particularly due to the current COVID-19 pandemic [6,11,12], but it had a negative impact on the early diagnosis and treatment of oncological disease [13]. Spain is among the leaders in European countries in active TD programs, with 25% of hospital services with some type of TD implemented in the consultation in 2014 [14], although there is still a long way to go as in the field of TD many innovative applications appear every year, with an increase expected in the future [8].

Many models exist for the application of TD. For example, the asynchronous or store and forward (S&F) method describes applications in which data are stored and retrieved independent of time and location, whereas the synchronous method or live interaction (LI) applications avail direct communication between the doctor and patient using videoconferences [15]. The hybrid forms combine S&F and LI, although given the greater administrative efforts of the live videoconferences, S&F is the most utilized method [16,17].

TD offers opportunities for improving access to medical care, the quality in care perceived, the satisfaction of the patients and health professionals [18,19], and the reduction in medical care costs [20]. Thus, TD offers some advantages over face-to-face consultations, reduces the waiting time and costs, improves access in rural areas, and/or for patients who have difficulties in going to face-to-face consultations, decreases the cancelations of appointments and absenteeism, and has been shown to be a useful tool for diagnosis and follow-up, favoring the early referral of patients with dermatological emergencies [6].

Although many economic assessments have been made [20], and many studies have been conducted on health results [18,19,21,22], many of them concluded that patients and dermatologists preferred face-to-face examinations. Additionally, no studies were found that delved into the experiences of individuals with the care received through TD. On the other hand, fluid, clear, and effective communication between the health personnel and the patient becomes even more important in remote health care, as it increases the satisfaction of the patients at the same time that it decreases their anxiety and uncertainty [23].

Thus, the objective of the article is to explore the experiences of the patients and the communication with health providers after 6 months of follow-up, by comparing TD versus Face-to-Face Dermatology (F-F/D).

## 2. Materials and Methods

### 2.1. Design

This study is part of a broader study on TD in the Poniente region of Almeria (Andalusia, Spain). A randomized, controlled, non-blinded study was conducted at various levels (Primary Care + Poniente University Hospital) and centers (all the health centers part of the Poniente Health District of Almeria), with 6 months of follow-up. The experimental group was composed of dermatology patients treated with remote communication technologies (TD), while the control group was composed of patients treated face-to-face at the hospital (F-F/D). The trial was registered at ClinicalTrials.gov ID: NCT04378296.

### 2.2. Participants and Setting

The study was conducted in the Poniente region of Almeria, with a reference population of 265,000 inhabitants. It was implemented in all the health centers that are a part of the Poniente Health District of Almeria and the Poniente University Hospital (El Ejido-Almeria). In 2021, approximately 9468 F-F/D consultations (first visits and revisions), and 9210 dermatology teleconsultations took place in the Poniente region of Almeria, Spain. The Clinical Management Units that were included in the study belonged to the Poniente Health District of Almeria, and were the following: Adra, Aguadulce, Alpujarra, Berja, Ejido Norte, Ejido Sur, El Parador, La Mojonera, Roquetas Norte, Roquetas Sur, Santa María, and Vícar.

The patients were randomly selected in both groups, with systematic random sampling for the selection of the group members. In this study, a random number was chosen, and inclusion into the group was proposed to one patient out of every three who entered the Primary Care consultation to request dermatological health care. In the case that the patient did not want to be included in the study, inclusion was proposed to the next patient.

To ensure that the sample was representative, the sample size was calculated for the original project and was based on the difference in means of the main variable with respect to the quality of life related with health measured through the EuroQol-5D. Assuming a two-tailed test with a level of significance of 0.05 and a power of 0.80, a sample size of 450 patients was established (225 for each tail) to detect a minimum difference of 0.04 points (from 0.90 to 0.86) in the index of utility of the EuroQol, between the experimental and control groups 6 months after the first visit. These values were adopted considering the reference population of the EuroQol for the Spanish population older than 18 years old (approx. 0.90), the clinically-relevant minimum difference in the EuroQol scale (0.04), and a standard deviation that was similar for both groups (0.15).

The number of participants included in each of the follow-up groups was proportionally distributed into each of the Clinical Management Units. Before the study, an analysis was conducted to verify the number of patients with dermatological problems who solicited these services according to health center. Afterwards, the proportion of each center with respect to the total was calculated.

The inclusion criteria were: (1) being older than 18 years old; (2) having a dermatological disease. To perform a real analysis of all the participants who entered the consultation and solicited health care, no dermatological pathologies were excluded; and (3) accepting to participate in the study. The exclusion criteria were: (1) non-dermatological pathology; (2) currently participating in another study; and (3) not wanting to participate in the study.

### 2.3. Measures and Instruments

The following sociodemographic and clinical characteristics of the participants were collected: age, gender, race/ethnicity, reason for consultation, diagnostic tests performed, anatomical location of the skin problem, treatment, number of primary care visits, and number of hospital visits. Additionally, an ad-hoc survey including items from the ‘Telehealth Patient Satisfaction Survey’ and a ‘costs survey’ were used. In this ad hoc survey, 8 items were utilized to discover the costs of both health care modalities, based on: (1) basic health zone; (2) time required for assistance; (3) transportation used; (4) patient’s employment status; (5) needs to be accompanied; (6) accompanying employment; (7) travel expenses; (8) calls to Primary Care/Hospital.

The evaluation of the experiences of the patients and communication was performed through an ad hoc questionnaire developed through the use of two validated tools. The questionnaire included the complete version of the Generic Short Patient Experiences Questionnaire (GS-PEQ) [24] and the adapted version of the Health Care Communication Questionnaire (HCCQ) [25]. The GS-PEQ is a short instrument that assesses the experiences of the patients with hospital care (see items 1–10). The objective of the HCCQ is to measure the communication experience of the ambulatory patients with the health professionals at the hospital (see items 11–17) (Table 1). This survey evaluates the care received by all healthcare professionals who, regardless of their professional category and/or follow-up alternative (TD or F-F/D), have had contact with patients.

### 2.4. Procedure

The nursing personnel from each of the Primary Care centers included in the study were in charge of informing and selecting the patients, as well as collecting the Clinical History data and providing the corresponding questionnaires. The patients were given two interviews, the first after being included in the study (month 0) and the second at the end of the follow-up period (after 6 months), at which time, the questionnaires were given to collect the data. The interviews were given in person during the visits scheduled or by telephone. During that period, patients had to attend both TD and F-F/D consultations, as many times as necessary depending on the severity of their pathology status.

All questionnaires were completed as follows:

Month 0: patients had to personally fill in the questionnaires after leaving the primary care office, in the waiting room. They then handed them to the nurse. If necessary, the nurse read the questions together with the response alternatives and the patient answered.

Month 6: in this case the following could happen:

(1) The patient’s dermatological problem has been solved before 6 months. In this case, the patient is called by telephone and the relevant questionnaires are administered.

(2) The patient’s dermatological problem has not been resolved before 6 months, and the patient does not have a scheduled hospital visit in month 6. In this case, a telephone call is made, and the pertinent questionnaires were administered;

(3) The patient’s dermatological problem has not been resolved before 6 months but the visit to the specialist coincides in month 6. In this case the patients will have to personally fill in the relevant questionnaires in the waiting room after leaving the specialist’s office. If necessary, the nurse will administer the questionnaires.

In cases where a telephone interview was necessary, optimal conditions were ensured for its performance, asking the patient for an environment where he could be calm and without interruptions.

Physicians and nurses were informed that a study was going to be developed in which the effectiveness of patients with dermatological problems would be analyzed. They would have to follow a protocol for the random selection of the patients, inform the patients, and request the informed consent form.

Patients were only aware of the information that appeared on the Patient Information Sheet and the Informed Consent Form. 

The protocol that was followed in the TD units (asynchronous modality) at the Poniente Health District of Almeria was the following:I.The patient went to the consultation of his or her Primary Care Physician (PCP) for a skin affliction, the PCP completed the teleconsultation form in the digital platform “Telederma,” indicating the affliction’s characteristics, localization, previous treatment (if any), background, etc. Afterwards, the PCP could directly give an appointment for the TD consultation or the patient could request it at the admissions desk at the primary health center;II.In a period under 10 working days, the patient attends the TD appointment in his or her primary care center. Upon entering, the nurse asks for the signed informed consent form that the PCP previously provided (the consent form is a document created by the Ministry of Health and Families from the Junta of Andalusia, through which the patient agrees to be attended telematically). Afterwards, the same nurse verifies that all the information needed by the dermatologist is correct, after which the images are taken using a set sequence according to the protocol, for their correct viewing. Independently of each of the lesions, a panoramic image is taken at a distance of at least 1 m in which the anatomical area is viewed clearly and posteriorly images with polarized light are taken with a dermatoscope; whenever possible, a measurement scale is used to determine the size of the lesion if a follow-up is needed. Then, the images are uploaded to the platform (one by one). On the upper part of the image, there is an “observations” section in which the incidences or observations can be described in detail, which could be useful for the dermatologist specialist at the hospital. Lastly, the images are eliminated from the camera before continuing with the next patient;III.In a period of a week, the patient goes back to his or her PCP to collect the results along with the diagnosis, treatment, and the indications deemed appropriate by the dermatologist, although if a severe pathology is suspected, the times are reduced to 1–2 days or less than 24 h.

### 2.5. Ethical Consideractions

The project was approved by the Research Ethics Committee from the Center-Almeria (Code CEIC-AL 27/2020). It was implemented according to the ethics and research guidelines found in the Declaration of Helsinki [26]. All the patients were provided with information about the study verbally and in writing, and were asked to sign an informed consent form before being included in the study. The appropriate measures needed to guarantee the privacy of the identification data of the patients were adopted, as established by the Organic Law of Protection of Personal Data 3/2018 [27]. 

### 2.6. Statistical Analysis

Analyses were carried out with SPSS 24th edition (SPSS Institute, Inc., Chicago, IL, USA) statistical software. Continuous variables were expressed as means with standard deviations (SDs), and categorical variables were presented as actual numbers and percentages. Patient baseline characteristics and potential differences between groups were compared using a difference in the mean values by employing a test for continuous variables and a difference in proportions test (binomial method), or the Chi-square test (replaced by Fisher’s exact test for cells with *n* < 5 cases) for qualitative variables. 

The questionnaire results are presented based on the comparison between the two groups, applying the Mann–Whitney U test for ordinal data and the chi-square test for nominal data. In addition, effect sizes were included. For interpreting the magnitude of effect sizes, Lovakov and Agadullina rules are followed [28]: r < 0.12—very small; 0.12 ≤ r < 0.24—small; 0.24 ≤ r < 0.41—moderate; and r ≥ 0.41—large.

## 3. Results

### 3.1. Patient Baseline Characteristics

The final sample after 6 months of follow-up was 450 participants (TD group: 225 versus F-F/D group: 225) (Figure 1) of which 53.3% were white women (92.2%), with a mean age of 52.16 year old (SD = 19.97 years). Baseline characteristics, depending on the intervention status are presented in Table 2. In general, the main reason for the consultation was a skin lesion (51.56%) localized on the head and neck (46.8%), followed by the legs (20.7%). For most of the cases, the treatment was pharmacological (54.7%), followed by surgical (27.3%). Statistically significant differences were not found between the TD and F-F/D groups with respect to age, race/ethnicity, anatomical location of the skin problem and calls to primary care/hospital. On the other hand, statistically significant differences were found in the variables reason for consultation, diagnostic tests performed (with more tests in the F-F/D group), treatment (more surgical treatments in the F-F/D group), number of primary care visits (more visits in the TD group), and number of hospital visits (more visits in the F-F/D group).

Most of the patients belonged to the basic health zones El Ejido (41.11%) and Roquetas de Mar (33.33%), in general requiring between 2–3 h (53.6%) and 1–2 h (36.4%) to go to the consultation at their health center/hospital, with 54.9% utilizing their own car for transport. As for the employment status of the patients, 34.7% were currently employed, 31.8% were retired, and 20.9% were unemployed. In addition, 63.1% of the patients needed to be accompanied to the consultation at the health center/hospital, and 83.6% had expenses associated to travel to the consultation. Lastly, most of the patients (81.1%) did not call their health center/hospital. Statistically significant differences were found between the TD and F-F/D groups with respect to basic health zones (with more patients in the TD group from areas further away from the hospital), time required for assistance (more time required to go to the consultation in the F-F/D group), transportation used (more use of public transport, their own car, taxis, and ambulance, for the F-F/D group), patient’s employment status, needs to be accompanied (greater need for being accompanied in the F-F/D group), and accompanying employment and travel expenses (more travel expenses in the F-F/D group) (Table 2).

### 3.2. Patients’ Experiences and Communication

With respect to the patients’ experiences and communication, the results from both of their follow-ups were in general positive, and as shown in Figure 2, statistically significant differences were found between the TD and F-F/D groups in questions 2, 3, 5, 6, 8, and 10 from the GS-PEQ, and questions 13–17 in the HCCQ (Appendix A).

As for the analysis of the items related with the patient’s experiences in both consultation modalities, the following was found:

Question 2, which enquired about confidence in the professional skills of the doctors, the TD group reported a greater confidence as compared to the F-F/D group (*p* < 0.01). In contrast, in question 3, which enquired if the participants received enough information about their diagnosis/affliction, the F-F/D group indicated having received more adequate information than the TD group (*p* < 0.01). However, the effect size of both questions was small or very small.

In questions 5 (which asked if the participants were involved in the decisions related with their treatment), and 8 (which asked if the help and treatment received was satisfactory), the F-F/D group obtained a higher score than the TD group (*p* < 0.01), with large effect sizes.

By contrast, in question 6 (which asked if the participants perceived that the work in the institution was well-organized), the TD group reported that the work was well organized, as compared to the F-F/D group (*p* < 0.01), with a moderate effect size.

Lastly, for question 10 (negative sense item which asked if the participants had received incorrect treatment), the F-F/D group obtained a lower score than the TD group (*p* < 0.01), with a large effect size.

As for the analysis of the items on communication perceived in both consultation modalities, we found the following:

In questions 13 (which asked about treatment with kindness), 14 (negative sense item that asked if they had been treated in a rude and hasty manner), 15 (which asked if the health professional addressed them with a smile), and 16 (if the health professional was able to adequately manage the consultation), the TD group obtained a better score in these items as compared to the F-F/D group (*p* < 0.01), with a moderate or large effect size in items 13, 15, and 16, and a very small effect size in item 14.

On the contrary, in question 17 (if the health professional showed respect for their privacy), the F-F/D group obtained higher scores than the TD group (*p* < 0.01), with a small effect size.

## 4. Discussion

The present study explored the experiences of the patients and the communication with the health professionals after a 6-month follow-up, comparing TD versus F-F/D. As far as we know, this is the first study that explores the experiences of the patients and communication with the health providers with respect to being treated remotely or not in a dermatology consultation.

The results revealed: (a) generalized positive experiences of the patients with both follow-up modalities; (b) significant differences about their diagnosis/affliction between the groups, in that TD patients received less information about their diagnosis/affliction as compared to the hospital consultation; they were less involved in the decisions related to their treatment, and less satisfied with the help and treatment received in the institution. However, the TD patients had more confidence on the professional skills of the doctors and perceived that the work in the institution was better organized; (c) significant differences in how the TD patients perceived better communication with the health professionals based on the treatment with kindness, addressing them with a smile, not being treated in a rude or hasty manner, and the ability of the professional to adequately manage the reason for the visit. However, the TD patients reported that they felt less respect for their privacy; (d) significant differences in how the F-F/D patients took longer to travel to the dermatology consultation at the hospital, and used their own vehicle as transport; and (e) that significant differences were not found between the follow-up groups in the rest of the items, for example, if the health professionals utilized technical words, perception on if the treatment was adapted to the situation of the patient, waiting time, or being treated in an aggressive manner.

The general positive experience found in our study in both follow-up groups is confirmed and in line with previous studies that analyzed experiences with telehealth devices in cardiology [29,30] or primary care [31,32], given that for the patients, remote communication with the health professionals was a medium that is flexible, convenient, easy to use, and acceptable for managing their health together. Although this study is the first that explores the experiences of the patients and the communication with health providers in both types of follow-up (TD vs. F-F/D), other studies have shown a high satisfaction with both follow-up modalities [18,19,21,33,34].

An important result in our study was that the patients with remote follow-up received less information about their diagnosis/affliction, were less involved in the making of decisions related with their treatment, and were less satisfied with the help and treatment received in person at the hospital. This is an important finding with respect to the expectations of the patients about the information provided by the health professionals. We believe that the TD patients had less possibilities of receiving answers from the health professionals about their doubts or questions associated to their diagnosis. Being well-informed, and having patient-centered care, is a key aspect in the subjective experiences of the patients, which are associated with a greater satisfaction [35]. Another study conducted with telemedicine, more specifically, in the remote follow-up of patients with pacemakers, also found that a characteristic aspect of the diagnostic process was that the patients experienced the feeling of not knowing or not being informed [36]. Although studies similar to the present one were not found, we believe that patients with remote follow-up have greater information needs than face-to-face follow-up, as they could have less opportunities to obtain information as compared to primary care at the hospital. In this sense, we coincide with Mason et al. [35] in that patient-centered care is one of the most important dimensions that has an effect on the satisfaction of telemedicine patients. A patient-centered focus could provide improvements in health care and in the clinical results in the remote follow-up of the patients.

Another important aspect from the results of the study that could be underlined was that patients with remote follow-up reported that telehealth professionals had better communication skills than the face-to-face hospital doctors. The telehealth providers need high-level communication skills to compensate for the lack of visual cues, for example [23]. Therefore, we can assume that the communication with the health professionals satisfied the needs of the patients in our study. These results are in agreement with other studies in which the analysis of the interaction between health professionals and patients demonstrated significant differences in communication between telemedicine and face-to-face meetings [37]. However, other studies have not shown differences between the communication of the health professionals and patients in either follow-up modalities (Telemedicine vs. in-person visits) [29,38]. Non-verbal communication is an important aspect of the interactions between health professionals and patients. Any effect of the non-verbal body language can decrease during remote meetings. Due to this, the health professionals in our study, in the TD follow-up group, developed better communication practices, as recommended by standards for an adequate communication in telemedicine [23,37].

Another result from our study that must be highlighted was that the TD patients reported feeling less respect for their privacy. This could be due to the use of images (photos and videos) during the examination and a lack of knowledge about what is done with that material after the consultation. The privacy of the patient could be at risk with this technology. Health professionals must obtain the patient’s consent before taking photos, and must explain how these will be used, utilizing adequate safety measures in their digital communications [39,40]. We believe that adequate information on the use of these images could contribute to the patients feeling safer, and that there was more respect for their privacy.

Lastly, statistically significant differences were found between the TD and F-F/D groups in the variables reason for consultation, diagnostic tests performed (with more tests in the F-F/D group), treatment (more surgical treatments in the F-F/D group), number of primary care visits (more visits in the TD group) and number of hospital visits (more visits in the F-F/D group). These results are consistent with what was expected to be found and with the bibliography found, since to carry out invasive diagnostic tests and surgical treatments, face-to-face care at the health center by a dermatologist is needed [15,41,42,43].

Despite the relevant results obtained, the present study has certain limitations. In the first place, some problems were found associated with the measurement of the patient’s experience, including a social desirability bias when intentionally answering the questionnaires to obtain positive results. Some patients were also confused because they believed that the questions were based on their experience with their state of health [44]. In the second place, this was a non-blinded study in which the doctors, researchers, and patients, knew the follow-up category of all the patients. In the third place, the size of the sample estimated in the original project was based on a main variable and not the individual items from the questionnaires utilized in this study. Nevertheless, we believe that this study has some important strengths, as it is a randomized study in a field of knowledge in which this type of design is not very frequent. This ensures a greater degree of evidence, a lower possibility of bias due to the random selection of the groups, and could be repeated and compared with other studies. Although there may have been a possible acquiescence bias in the study, an attempt has been made to control using scales and multiple-choice questions with some items worded negatively, to avoid the response tendency. On the other hand, during the interviews, patients insisted on being honest in their answers. On the other hand, the study provides data about an area in which research is lacking, providing critical data that could be used in the successful implementation of new eHealth services. This could help to overcome the barriers found for the successful implementation of new eHealth services and health care models in which the experiences of and communication with the patients are crucial to ensure patient-centered care [35,45].

The results can be used by public health service managers to further investigate patients’ experiences and communication in areas not included in this study. In the future, it might be useful to record and share with the reader the reasons why patients may refuse to receive health care via teledermatology and thus be excluded from the study. Additionally, as the patient–doctor interaction is a mutual relationship, it would be interesting to collect data on the physicians’ experience with patients through teledermatology.

## 5. Conclusions

The results from the study showed that the experiences of the patients and the communication with health providers after 6 months of follow-up, when comparing follow-up through TD versus F-F/D, were generally positive. 

As for the patient’s experiences, the TD group perceived less information about their diagnosis, were less involved in the decisions, and were less satisfied with the help and treatment received in the institution. However, the patients who received follow-up through TD had more confidence in the professional skills of the doctors and believed that the work at the institution was better organized.

Lastly, the patients from the TD group perceived better communication skills of the health professionals, although they also felt that less respect was given to their privacy.

This study provides important information about the patient’s experiences and communication, and although it confirms positive communication experiences, there is a great need to improve specific areas in which the users reported negative experiences in the TD group. These actions will help to create better eHealth services focused on the patient.

## Figures and Tables

**Figure 1 jcm-11-05528-f001:**
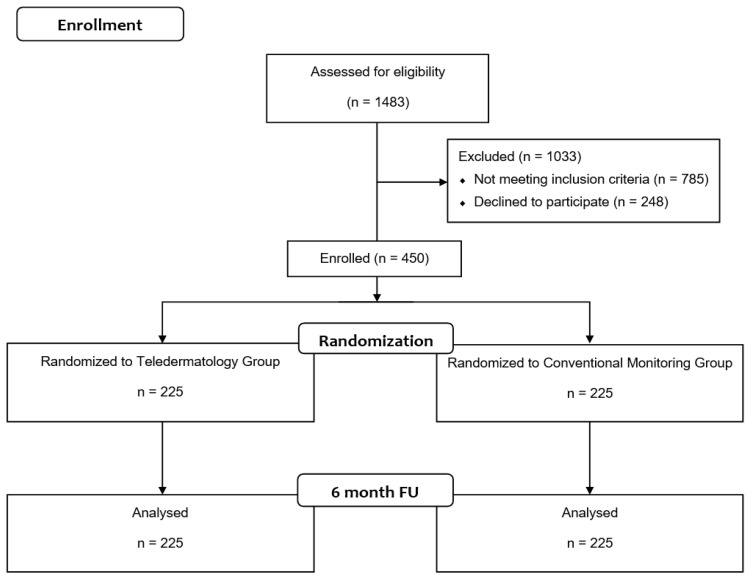
CONSORT Flow Diagram.

**Figure 2 jcm-11-05528-f002:**
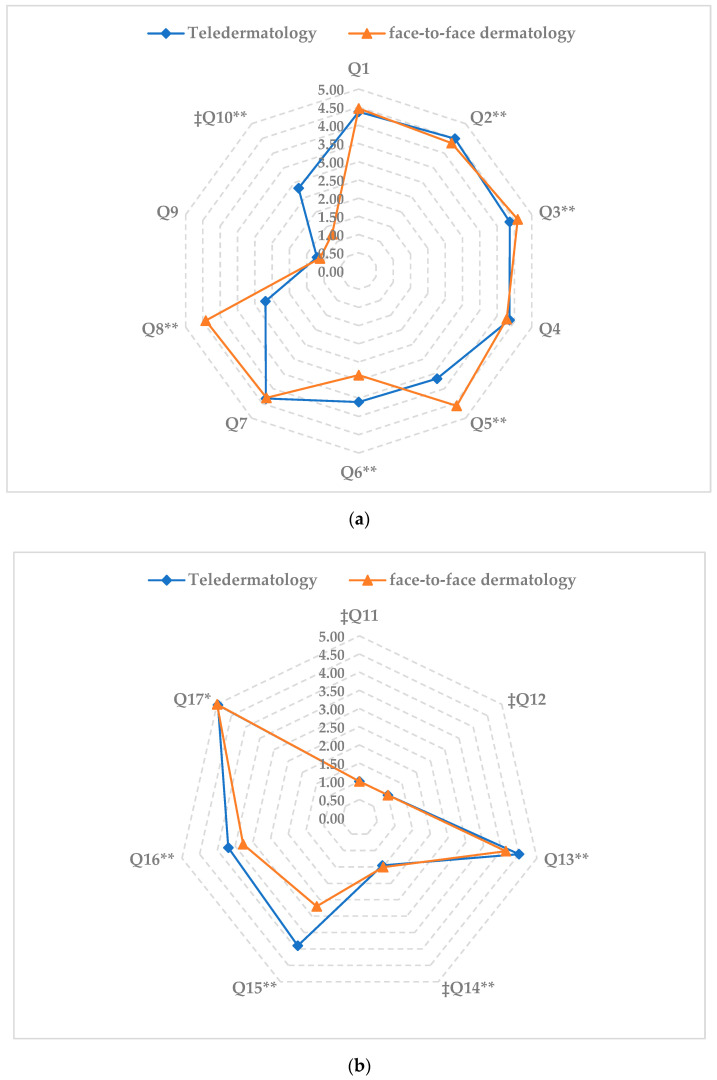
Results derived from patient experiences and communication questionnaire. (**a**) Patient experiences items; ** *p* < 0.01; ‡ negative items; see Table 1 for the content of the items. (**b**) Communication items; ** *p* < 0.01, * *p* < 0.05; ‡ negative items; see Table 1 for the content of the items.

**Table 1 jcm-11-05528-t001:** Patient experiences and communication questionnaire.

Items	Possible Answers
Q1. Did the clinicians talk to you in a way that was easy to understand?	1 = Not at all; 2 = To a small extent; 3 = To a moderate extent; 4 = To a large extent; 5 = To a very large extent
Q2. Do you have confidence in the clinicians’ professional skills?	
Q3. Did you get sufficient information about your diagnosis/afflictions?	
Q4. Did you perceive the treatment as adapted to your situation?	
Q5. Were you involved in decisions regarding your treatment?	
Q6. Did you perceive the institution’s work as well organized?	
Q7. Did you have to wait before you were admitted for services at the institution?	
Q8. Overall, was the help and treatment you received at the institution satisfactory?	
Q9. Overall, what benefit have you had from the care at the institution?	
Q10. Do you believe that you were in any way given incorrect treatment? *	
Q11. I was asked questions in an aggressive manner *	1 = Nothing; 2 = A little; 3 = Quite a bit; 4 = A lot; 5 = Very much
Q12. I was given answers in an aggressive manner *	
Q13. I was treated with kindness	
Q14. I was treated in a rude and hasty manner *	
Q15. The healthcare provider addressed me with a smile	
Q16. The healthcare provider was able to manage the consultation	
Q17. The healthcare provider showed respect for my privacy	

* Negative items.

**Table 2 jcm-11-05528-t002:** Selected patient baseline characteristics.

	All	Groups		*p*-Value
		Teledermatology (*n* = 225)	Face-to-Face Dermatology (*n* = 225)	
Age M(SD)	52.16 (19.97)	52.53 (18.17)	51.78 (21.65)	0.744 ^a^
Women *n* (%)	240 (53.3%)	131 (58.2%)	109 (48.4%)	0.047 ^b^
Race/ethnicity *n* (%)				
White/Caucasian	415 (92.22%)	208 (94.4%)	207 (92%)	0.453 ^b^
Gypsy	6 (1.33%)	4 (1.78%)	2 (0.89%)	
Hispanic/Latino	10 (2.22%)	5 (2.22%)	5 (2.22%)	
Black	3 (0.67%)	0 (0%)	3 (1.33%)	
Arab	16 (3.56%)	8 (3.56%)	8 (3.56%)	
Reason for consultation *n* (%)				
Injury	232 (51.56%)	140 (62.22%)	92 (40.89%)	0.000 ^b^
Rash	52 (11.56%)	24 (10.67%)	28 (12.44%)	
Injury and rash	14 (3.11%)	8 (%)	6 (2.67%)	
Others *	152 (33.78%)	53 (23.56%)	99 (44%)	
Diagnostic tests performed *n* (%)				
None	363 (80.67%)	211 (93.78%)	152 (67.56%)	0.000 ^b^
Blood test	31 (6.89%)	7 (3.11%)	24 (10.67%)	
Biopsy	52 (11.56%)	6 (2.67%)	46 (20.44%)	
Micro punctures	3 (0.67%)	1 (0.44%)	2 (0.89%)	
Anatomical location of the skin problem *n* (%)				
Trunk	83 (18.5%)	41 (18.2%)	42 (18.8%)	0.336 ^b^
Limbs	93 (20.7%)	42 (18.7%)	51 (22.8%)	
Head and neck	210 (46.8%)	113 (50.2%)	97 (43.3%)	
Limbs. head and neck	13 (2.9%)	6 (2.7%)	7 (3.1%)	
Trunk and limbs	24 (5.3%)	14 (6.2%)	10 (4.5%)	
Whole body	10 (2.2%)	5 (2.2%)	5 (2.2%)	
Trunk. head and neck	16 (3.6%)	4 (1.8%)	12 (5.4%)	
Treatment *n* (%)				
Pharmacological	246 (54.7%)	146 (64.9%)	100 (44.4%)	0.000 ^b^
Surgical	123 (27.3%)	31 (13.8%)	92 (40.9%)	
Follow-up and evolution	55 (12.2%)	45 (20%)	10 (4.4%)	
Pharmacological. surgical and follow-up	26 (5.8%)	3 (1.3%)	23 (10.2%)	
Number of Primary Care visits M (SD)	1.96 (0.76)	2.24 (0.65)	1.68 (0.76)	0.000 ^a^
Number of hospital visits M (SD)	0.75 (0.95)	0.01 (0.09)	1.48 (0.85)	0.000 ^a^
Basic health zones *n* (%)				
El Ejido	185 (41.11%)	122 (54.22%)	63 (28%)	0.000 ^b^
Adra	46 (10.22%)	29 (12.89%)	17 (7.56%)	
Vícar	35 (7.78%)	9 (4%)	26 (11.56%)	
Roquetas de Mar	150 (33.33%)	62 (27.56%)	88 (39.11%)	
Berja	34 (7.56%)	3 (1.33%)	31 (13.78%)	
Time required for assistance *n* (%)				
<1 h	15 (3.3)	15 (6.7)	0 (0)	0.000 ^b^
1–2 h	164 (36.4)	160 (71.1)	4 (1.8)	
2–3 h	241 (53.6)	49 (21.8)	192 (85.3)	
3–4 h	30 (6.7)	1 (0.4)	29 (12.9)	
>4 h	0 (0)	0 (0)	0 (0)	
Transportation used *n* (%)				
Public transport	39 (8.7)	16 (7.1)	23 (10.2)	0.000 ^b^
Own car	247 (54.9)	97 (43.1)	150 (66.7)	
Ambulance	51 (11.3)	17 (7.6)	34 (15.1)	
Taxi	9 (2)	3 (1.3)	6 (2.7)	
Other	104 (23.1)	92 (40.9)	12 (5.3)	
Patient’s employment status *n* (%)				
Working	156 (34.7)	89 (39.6)	67 (29.8)	0.005 ^b^
Unemployed	94 (20.9)	53 (23.6)	41 (18.2)	
Pensioner	143 (31.8)	66 (29.3)	77 (34.2)	
Sick leave	11 (2.4)	4 (1.8)	7 (3.1)	
Other	46 (10.2)	13 (5.8)	33 (14.7)	
Needs to be accompanied *n* (%)				
No	166 (36.9)	100 (44.4)	66 (29.3)	0.001 ^b^
Yes	284 (63.1)	125 (55.6)	159 (70.7)	
Accompanying employment *n* (%)				
Working	72 (24.9)	30 (22.7)	42 (26.8)	0.022 ^b^
Unemployed	140 (48.4)	57 (43.2)	83 (52.9)	
Pensioner	67 (23.2)	37 (28)	30 (19.1)	
Sick leave	4 (1.4)	2 (1.5)	2 (1.3)	
Other	6 (2.1)	6 (4.5)	0 (0)	
Travel expenses *n* (%)				
No	74 (16.4)	51 (22.7)	23 (10.2)	0.000 ^b^
Yes	376 (83.6)	174 (77.3)	202 (89.8)	
Calls to Primary Care/Hospital *n* (%)				
Never	365 (81.1)	183 (81.3)	182 (80.9)	0.988 ^b^
1	49 (10.9)	24 (10.7)	25 (11.1)	
2	36 (8)	18 (8)	18 (8)	
More than 2 times	0 (0)	0 (0)	0 (0)	

^a^ t-Student test; ^b^ Chi Squared test; * The reason “Others” includes several dozens of different dermatologic pathologies that alone do not reach 2%. Because of the above, it was decided to include in the same group all dermatologic pathologies that did not reach at least 2%.

## Data Availability

The datasets used and/or analyzed during the current study are available from the corresponding author on reasonable request.

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
