# Peer review of "Patients’ Experiences and Communication with Teledermatology versus Face-to-Face Dermatology"

_jcm, 2022, doi:10.3390/jcm11195528_

Round 1
Reviewer 1 Report
The authors explore and try to answer questions about the yet to be addressed problem of patient perception and satisfaction when comparing teledermatology and face to face dermatology. The topic is appropriate in a time in which many services and businesses have increased their ability to provide people with solutions and goods without face to face interactions. On the other hand, healthcare services are still providing a small portion of doctor-patient interaction through telemedicine. Before going through with making the availability of teledermatology wider and its use more frequent or even predominant, it is fair and necessary to investigate the implications and downsides of it, since it definitely has many differences with face to face doctor-patient interaction. Among these, it is fair to wonder if patients perceive physicians, disease, treatment and empathy in the same way through teledermatology.
Easy to understand and concise language is used in the article. Materials and methods are described thoroughly. In order to have a greater impact on improving teledermatology in the future, it might have been useful to record and share with the reader reasons why patients refused to receive healthcare assistance through teledermatology and were therefore excluded from the study. Also, being the patient-doctor interaction in both directions, it would be interesting to collect data on the physicians’ experience with patients through teledermatology. These two options might not be feasible for this study anymore, but could be hinted at in the discussion in order to guide future studies in the field.
Charts, questionnaires and graphs seem appropriate, useful and well presented. It would be useful to highlight in some way the “negative items”, as defined by the author, in the figure 2 graphs.
The article is overall detailed and the study has been carried out with a method appropriate to the subject. It is also one of the first to tackle the problem, and probably the very first one to bring in depth exploration of it.
Finally, we must state that teledermatology has been implemented due to the pandemic [Nazzaro G. What is the role of a dermatologist in the battle against COVID-19? The experience from a hospital on the frontline in Milan. Int. J. Dermatol. 2020;59:e238–e239] but it had a negative impact on the early diagnosis and treatment of oncological diseases [Trepanowski N., Chang M.S., Zhou G., Ahmad M., Berry E.G., Bui K., Butler W.H., Chu E.Y., Curiel-Lewandrowski C., Dellalana L.E., et al. Delays in melanoma presentation during the COVID-19 pandemic: A nationwide multi-institutional cohort study. J. Am. Acad. Dermatol. 2022;20:01023–01024]
Reviewer 2 Report
Please address the following points points:
To what extent were patients, physicians, interviewers aware of the study?
Could the nurse who did the initial intake of the patient have an influence on the patient’s perception and response?
The mean age is quite old, there could be a large differences in technology/computer ability could this have influenced the result?
If the patients were randomly assigned why were TD patients living further away, shouldn’t this be included in the initial baseline demographic?
The “other” reason for consultation makes up a great proportion of both the TD and FFD groups, why wasn’t it further disambiguated?
The figure illustrating patient experience items is difficult to understand without a legend, would recommend also providing TD an FFD in a tabular form OR adding a brief summary to the figure. E.g. instead of Q2, Confidence in Professional Skill
Should have described the setting for the TD appointments on the phone. Were there conditions enforced such as the patient is not actively participating in any activity or the patient is in a private area?
In the discussion should include more literature review e.g. why do FFD appointments lead to more invasive procedures?
How did the patients complete the 17 item questionnaire? Did the interviewer ask the patient the questions? Was it an electronic phone survey? any possibility of acquiescence bias?
Did patients have additional visits in between 0 and 6 months?
The survey’s wording of “healthcare provider” and “clinician” does that refer to all members of the health care team, nurse, PCP, or dermatologist?
Round 2
Reviewer 2 Report
Authors have made significant revisions to the paper that improved its quality.